# Transcranial Direct Current Stimulation in Episodic Migraine: A Systematic Review and Meta-Analysis of Randomized Controlled Trials [note 1]

**DOI:** 10.3390/medsci13030084

**Published:** 2025-06-26

**Authors:** Faraidoon Haghdoost, Abdul Salam, Fatemeh Zahra Seyed-Kolbadi, Deepika Padala, Candice Delcourt, Anthony Rodgers

**Affiliations:** 1The George Institute for Global Health, University of New South Wales (UNSW), Sydney, NSW 2000, Australia; asalam@georgeinstitute.org.in (A.S.); cdelcourt@georgeinstitute.org.au (C.D.); arodgers@georgeinstitute.org (A.R.); 2The George Institute for Global Health, Hyderabad 500034, India; dpadala@georgeinstitute.org.in; 3Prasanna School of Public Health, Manipal Academy of Higher Education (MAHE), Manipal 576104, India; 4Evidence Based Medicine Center, Hormozgan University of Medical Sciences, Bandar Abbas 79196-93116, Iran; fatemehzahraseyedkolbadi@gmail.com; 5Department of Clinical Medicine, Faculty of Medicine, Health and Human Sciences, Macquarie University, Sydney, NSW 2113, Australia

**Keywords:** brain stimulation, headache, migraine, neuromodulation, prophylaxis, tDCS

## Abstract

**Background:** Transcranial direct current stimulation (tDCS) is a non-invasive neuromodulation technique for migraine prevention. This study evaluates the efficacy of tDCS compared to sham in preventing episodic migraine in adults. **Methods:** PubMed and Embase databases were searched until May 2025 to identify randomized controlled trials comparing tDCS with sham for the prevention of episodic migraine in adults. Risk of bias in the included trials was assessed using the Cochrane Risk of Bias Tool version 2. A random effect meta-analysis was conducted to evaluate the effects of cathodal and anodal tDCS on migraine frequency (days per month and attacks per month). **Results:** The meta-analysis included six trials with 172 participants (mean age 34 years, 82% females). Both cathodal (three studies, over the occipital area) and anodal (three studies, over the occipital or primary motor area) tDCS reduced the mean number of monthly migraine days and migraine attacks compared to sham. After pooling the outcomes and excluding two studies at high risk of bias, anodal tDCS over the occipital or primary motor area (standardized difference in means = −0.7, 95% CI: −1.7, 0.2, *p* = 0.124) and cathodal tDCS over the occipital area (standardized difference in means = −0.7, 95% CI: −1.1, −0.3, *p* = 0.000) reduced headache frequency compared to sham. However, the reduction with anodal tDCS was not statistically significant. **Summary:** tDCS may be effective in preventing episodic migraine. However, the evidence is limited by the small number of heterogeneous trials, with variation in electrode placement and stimulation intervals.

## 1. Introduction

Migraine is the leading cause of disability in young women and the second leading cause of disability worldwide [1,2]. The International Classification of Headache Disorders, 3rd edition (ICHD-3), defines migraine as a headache lasting 4 to 72 h with at least two of these four features: unilateral pain, a pulsating nature, moderate to severe intensity, and aggravation by physical activity [3]. The headaches should also present with the following: nausea (and/or vomiting), and/or sensitivity to light (photophobia), and sensitivity to sound (phonophobia) [3]. Migraine is classified into two types based on the presence or absence of aura: migraine with aura and migraine without aura [3]. Migraine with aura involves transient, fully reversible neurological symptoms—most commonly visual or sensory disturbances—that usually develop gradually over minutes and are typically followed by headache and other migraine-related features [3]. Additionally, migraine is categorized as episodic or chronic based on the number of headache days per month [3]. Chronic migraine is defined as having headaches on 15 or more days per month for more than three months, with at least eight of those days featuring migraine headache characteristics [3]. In contrast, when headache occurs on fewer than 15 days per month, it is classified as episodic [3].

Preventive interventions are required for migraine patients with ≥2 migraine days per month related to a reduced quality of life [4,5]. Migraine prevention includes pharmacological and non-pharmacological approaches [6,7]. Some antihypertensive medications such as beta-blockers and sartans like candesartan, as well as anticonvulsants, tricyclic antidepressant, botulinum toxin and anti-calcitonin gene-related peptide (anti-CGRP) medications are among the suggested medications for the prevention of migraine [6,8].

Given the significant burden of migraine and the limitations of pharmacological treatments—such as adverse effects, lack of efficacy in some patients, and contraindications in others—non-invasive brain neuromodulation (NIBS) approaches have emerged as promising non-pharmacological alternatives for migraine prevention. These include transcranial magnetic stimulation (TMS), transcutaneous vagus nerve stimulation, transcutaneous supraorbital stimulation, transcranial alternating current stimulation (tACS), transcranial random noise stimulation (tRNS) and transcranial direct current stimulation (tDCS), some of which are already available in certain countries [7,9,10]. Particularly for people intolerant to pharmacological treatments, NIBS methods have the potential to be an effective and a safe alternative in preventing migraine [9,11]. In addition to neuromodulation, lifestyle-based interventions—such as telecoaching to manage physical activity—and behavioral treatments have also shown potential as non-pharmacological strategies for migraine prevention [7,12].

tDCS has two electrodes producing a constant low-intensity current flow from the anode (excitatory) to the cathode (inhibitory), altering cortical excitability by depolarizing or hyperpolarizing underlying neurons, respectively [13,14]. Electrode placement can be cephalic (on the head) or extracephalic (for example, on the arms), and stimulation can be delivered in single or dual configurations, depending on the target area and study design [15,16,17]. Stimulation may also be applied during a task (online) or before it (offline), as the chosen protocol can influence clinical and behavioral outcomes [18]. Previous studies have reported that both cathodal and anodal tDCS are effective and safe for migraine prevention [19,20]. However, a systematic review and meta-analysis is needed to synthesize the available evidence from randomized controlled trials. This systematic review and meta-analysis aimed to evaluate the efficacy of tDCS compared to sham in preventing episodic migraine, and to compare the effects of anodal versus cathodal lead placement.

## 2. Method

This systematic review and meta-analysis of randomized clinical trials (RCT)s was registered on the International Prospective Register of Systematic Reviews (PROSPERO) with registration number CRD42020215735 [21] and is reported according to Preferred Reporting Items for Systematic Reviews and Meta-analyses (PRISMA) [22].

### 2.1. Search Strategy

PubMed and Embase (via Ovid) databases were systematically searched until 9 May 2025. The final search results and the search strategy are reported in Appendix A.

### 2.2. Inclusion and Exclusion Criteria

We included RCTs evaluating the effect of tDCS compared to sham in participants with episodic migraine. Episodic migraine was defined based on the International Classification of Headache Disorders (ICHD) as experiencing headache less than 15 days per month [3]. The details of the inclusion criteria are reported in Table 1. Only published studies in English language were included. Conference abstracts, open-label studies and letters to the editors were excluded.

### 2.3. Study Selection and Data Extraction

Citations were downloaded from the databases and imported into Rayyan, an online free platform for performing systematic reviews [23]. After removing the duplicates, citations were screened based on their title and abstract by two reviewers in duplicate (by FH and DP), independently. Then, included references were screened in duplicate (by FH and DP), independently, based on the full text. Conflicts between the reviewers including or excluding studies, we resolved via discussion or with the help of a senior reviewer (AS). Independent reviewers (FH, DP, FZSK) performed data extraction in duplicate using a predesigned Microsoft Excel spreadsheet. Any disagreements were resolved by discussion and if needed, with the help of a senior investigator (AS).

### 2.4. Risk of Bias Assessment

Version 2 of the Cochrane risk-of-bias tool (RoB 2) for randomized trials was used for assessing risk of bias in the included RCTs [24]. Five main domains were evaluated for RoB, including ‘bias arising from the randomization process’, ‘bias due to deviations from intended interventions’, ‘bias due to missing outcome data’, ‘bias in measurement of the outcome’ and ‘bias in selection of the reported result’. One extra domain for evaluating “Bias arising from period and carryover effects” was assessed for cross-over studies. Independent reviewers assessed the RoB in duplicate (FH, DP, FZSK). A senior reviewer (AS) was involved in case of any conflict. Overall RoB for each included paper was reported as ‘low risk of bias’, ‘some concerns’ or ‘high risk of bias’.

### 2.5. Outcomes

The only dependent variables measured were the number of migraine-related days in a month and the frequency of migraine attacks, and a composite outcome combining both.

### 2.6. Data Analysis

Descriptive data were reported as mean (of days or attacks per month) ± standard deviation (SD) or number (%). When a variable was reported by two or more studies a meta-analysis was conducted. Differences in means and 95% confidence intervals (95% CI) were reported as the effect size using the DerSimonian and Laird random-effect model [25]. To pool studies that have reported different headache frequency outcomes, we reported standardized difference in means and 95% CI. For sensitivity analysis, we excluded studies with “high risk” of bias. Statistical heterogeneity was assessed by reporting I^2^ and *p*-value statistics. *p*-value < 0.05 was considered as presence of statistical heterogeneity among the RCTs. I^2^ values under 40% were regarded as having no heterogeneity, 40% to 70% as having moderate and over 75% as having significant heterogeneity [26]. Comprehensive Meta-Analysis (CMA), version 4 (Biostat, Englewood, NJ, USA) was used to analyse the data.

## 3. Results

In total, 314 references from PubMed (n = 86) and Embase (n = 228) were found. After screening, eight RCTs were included (Figure 1 and Table 2). However, two studies were excluded from the meta-analysis. The study by Hasirci Bayir et al. [27] was excluded because (a) it was unclear whether the participants had episodic migraine, chronic migraine, or a mix of both; and (b) standard deviations or standard errors were not reported in the figures or manuscript. Rather than estimating the missing data, we chose to exclude this study from the meta-analysis. In addition, the study by DaSilva et al. [28] was excluded because (a) it used high-definition tDCS rather than the conventional tDCS used in all other included studies, and (b) it reported days with moderate or severe headache as the outcome, instead of overall headache or migraine days.

A total of six RCTs with 172 participants were included. The mean age of participants was 34 ± 6 years, and 82% were female. RCTs were published between 2011 and 2024, and five (83%) had parallel designs. Four (67%) RCTs were assessed as having “some concerns”, and two (33%) studies as having “high risk” of bias (Figure 2).

Three studies used cathodal tDCS. The cathode electrode was placed over the occipital (Oz), and the other electrode was placed over the vertex (Cz) or supraorbital area (Figure 3). Anodal tDCS was used in three studies: anode electrode was placed over the occipital (Oz) or primary motor (M1) cortex, and the other electrode was placed over the vertex (Cz) or supraorbital area (Figure 3).

Participants receiving anodal tDCS over the primary motor area had lower mean monthly migraine attacks (two studies, difference in means = −1.0, 95% CI: −1.6, −0.5, I-squared = 0.00) than sham (Figure 4). Only one anodal study (over the occipital area) reported monthly migraine days that showed 1.9 days reduction (95% CI: −3.1, −0.7) compared to sham. Participants on cathodal tDCS over the occipital area (two studies) had 4.2 days less monthly migraine (95% CI: −7.2, −1.2, I-squared = 34.18) compared to sham. Cathodal tDCS (over the occipital area) reduced 0.3 migraine attacks per month compared to sham (two studies, 95% CI: −1.3, 0.7, I-squared = 0.00) (Figure 4). The degree of heterogeneity within each group was not considerable.

By reporting standardized difference in mean, the two outcomes (migraine days and migraine attacks per month) were pooled together (composite outcome) for further analysis. Anodal tDCS over the occipital or primary motor area (standardized difference in mean = −0.9, 95% CI: −1.5, −0.3, *p* = 0.005, I-squared = 47.83) and cathodal tDCS over the occipital area (standardized difference in mean = −0.5, 95% CI: −1.0, 0.0, *p* = 0.051, I-squared = 33.53) reduced migraine frequency compared to sham (Figure 5). After removing the studies with a high risk of bias, both types of tDCS reduced migraine frequency (cathodal was significant) compared to sham. (Figure 6).

## 4. Discussion

Compared to sham, cathodal tDCS over the occipital area and anodal tDCS over the occipital or primary motor area reduced the number of monthly migraine days and monthly migraine attacks in participants with episodic migraine. Six studies comprising 172 participants were included in the meta-analysis, and they were associated with either a high risk of bias or some concerns (none had a low risk of bias). The included publications also used various lead placement techniques. To overcome this limitation, we divided the studies into two groups of cathodal and anodal. Also in the sensitivity analysis, we eliminated trials with a high risk of bias and discovered that both types of tDCS, with the cathodal type being significant, reduced migraine frequency outcomes compared to sham.

Previous reviews have found similar results to ours that cathodal and anodal tDCS are both beneficial for preventing migraine [34,35,36,37,38]. In our review, studies on cathodal tDCS aimed occipital cortex and trials on anodal tDCS targeted the occipital or primary motor cortex. None of the included anodal RCTs in the current analysis focused on dorsolateral prefrontal cortex (DLPFC) which is mentioned in the literature [34]. There are not many studies that have compared various lead positions to one another. In a comparison, anodal tDCS over the DLPFC had better outcome than primary motor cortex and sham in participants with refractory chronic migraine. However, the study’s sample size was only 13 people [39]. Another study evaluating different lead placements in chronic migraine suggested that the most effective outcomes were observed with specific multi-montage protocols rather than individual anodal or cathodal placements alone [40]. In particular, protocol one—which combined anodal tDCS over the right ventrolateral prefrontal cortex and cathodal over the left dorsomedial and superior frontal gyrus, followed by anodal stimulation of the right primary motor cortex and cathodal stimulation over the medial crosstalk of the hemispheres—demonstrated the highest effect size. This nuanced combination highlights the importance of montage integration and sequence. Therefore, while these findings align partially with earlier hypotheses about effective sites, the diversity in protocol design, polarity, and targeted networks contributes to substantial inter-protocol variability. As such, it remains challenging to define a single optimal technique or lead placement, and individualized approaches may be necessary to maximize therapeutic efficacy [34,40].

### 4.1. Strengths, Limitations and Future Directions

One limitation of our review is that we only searched two databases, PubMed and Embase. Although we believe this strategy captured the majority of relevant studies, a more comprehensive search, including other databases and grey literature, may identify additional evidence. We also included only studies published in English and excluded conference abstracts, preferring to focus on peer-reviewed full-text articles. This approach may have led to the omission of potentially relevant studies published in other languages or presented only in abstract form.

We also focused exclusively on studies involving episodic migraine. While this focus allowed us to reduce heterogeneity and ensure comparability of included studies, it led to the exclusion of otherwise relevant trials, such as that by Rahimi et al. [41], which included both episodic and chronic migraine patients without reporting separate data. Nonetheless, this focus can also be considered a strength, as episodic and chronic migraine may respond differently to neuromodulation interventions.

The available controlled clinical trials had small sample sizes, utilized various lead placement, study duration, and intervention intervals, and evaluated various outcomes. There is a need for further research with standardized designs that adhere to generally acknowledged guidelines for migraine trials especially regarding study duration and outcome assessment [42]. Large-scale controlled clinical trials with multiple arms are required to compare the effect of various tDCS lead locations on migraine prevention. Ornello et al. [34] have also suggested performing long-term multicenter trials, with evaluating more outcomes, and considering comorbidities and migraine characteristics.

### 4.2. Conclusions

Both cathodal and anodal tDCS appear to be potentially effective preventive treatments for episodic migraine. However, further research is needed to confirm these findings, as the currently available studies have small sample sizes and employ varied lead placement methods, making it difficult to recommend an optimal approach.

## Figures and Tables

**Figure 1 medsci-13-00084-f001:**
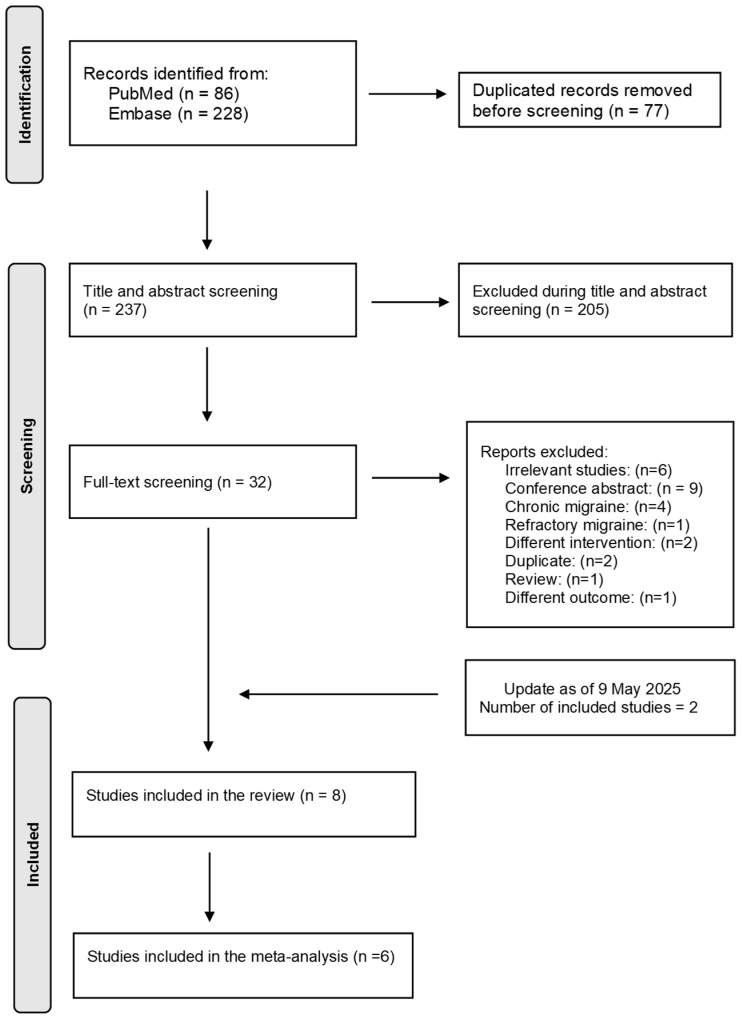
Flow diagram.

**Figure 2 medsci-13-00084-f002:**
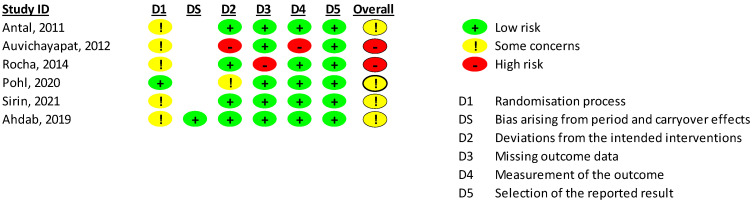
Risk of bias assessment for included studies. Ahdab 2019 [20], had a cross-over, and the other studies had parallel designs. Antal, 2011 [29], Auvichayapat, 2012 [30], Rocha, 2014 [31], Ahdab, 2019 [20], Pohl, 2020 [32], Sirin, 2021 [33].

**Figure 3 medsci-13-00084-f003:**
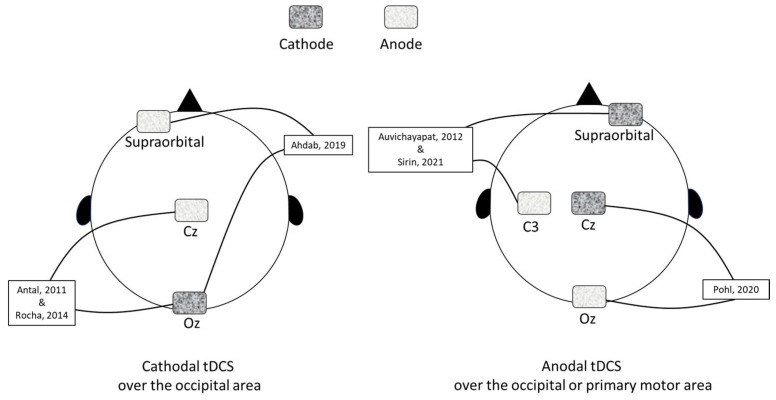
Cathodal and anodal tDCS lead placements in the included studies in the meta-analysis. Refer to Table 1 for more details. Antal, 2011 [29], Auvichayapat, 2012 [30], Rocha, 2014 [31], Ahdab, 2019 [20], Pohl, 2020 [32], Sirin, 2021 [33].

**Figure 4 medsci-13-00084-f004:**
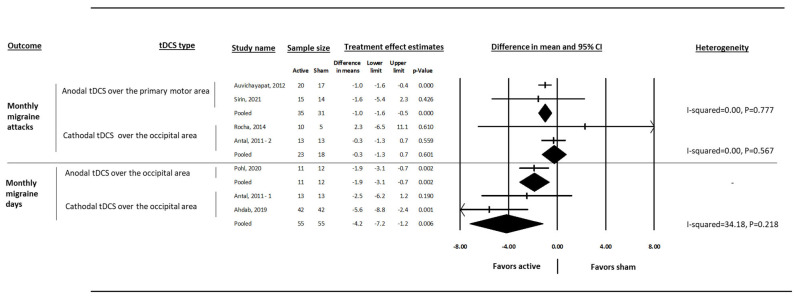
The effect of cathodal and anodal tDCS on the difference in means of monthly migraine attacks and monthly migraine days in people with episodic migraine. Antal, 2011 [29], Auvichayapat, 2012 [30], Rocha, 2014 [31], Ahdab, 2019 [20], Pohl, 2020 [32], Sirin, 2021 [33].

**Figure 5 medsci-13-00084-f005:**
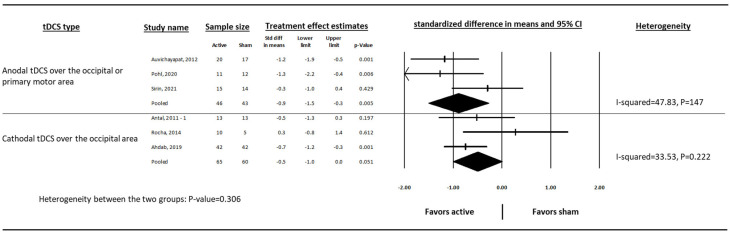
The effect of cathodal and anodal tDCS on the standardised difference in means of monthly migraine frequency in people with episodic migraine. Standardized mean difference is a unitless measure that expresses the size of the effect in terms of standard deviations. It allows comparison across studies even when the frequency outcomes were measured using different scales. A higher absolute value indicates a stronger effect. Antal, 2011 [29], Auvichayapat, 2012 [30], Rocha, 2014 [31], Ahdab, 2019 [20], Pohl, 2020 [32], Sirin, 2021 [33].

**Figure 6 medsci-13-00084-f006:**
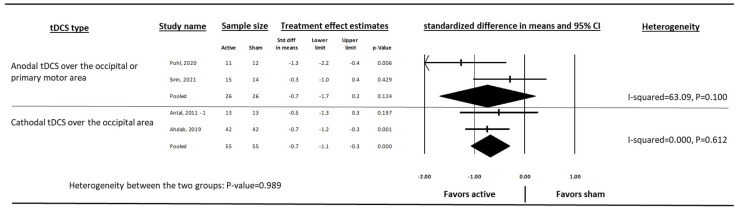
The effect of cathodal and anodal tDCS on the standardised difference in means of monthly migraine frequency in people with episodic migraine after excluding studies with a high risk of bias. Standardized mean difference is a unitless measure that expresses the size of the effect in terms of standard deviations. It allows comparison across studies even when the frequency outcomes were measured using different scales. A higher absolute value indicates a stronger effect. Antal, 2011 [29], Ahdab, 2019 [20], Pohl, 2020 [32], Sirin, 2021 [33].

**Table 1 medsci-13-00084-t001:** PICOs criteria for including randomised clinical trials in the current systematic review.

PICOs	Explanation
Population (P)	Adult participants (age ≥ 18 years) with episodic migraine
Intervention (I)	Transcranial direct current stimulation (both cathodal and anodal tDCS)
Comparator (C)	Sham *
Outcome (O)	Difference in change from baseline in migraine days per month and migraine attacks per month
Study design (s)	Randomized controlled trials

* In clinical studies, particularly those involving physical or procedural interventions, sham treatments are used as a control condition to mimic the active intervention without delivering its therapeutic component. This design helps isolate the specific effects of the intervention from nonspecific influences such as the placebo effect, which includes psychological or physiological responses to receiving treatment.

**Table 2 medsci-13-00084-t002:** Characteristics of included studies.

Study Name	Country	Sample Size, Female (%), Age (Mean)	Reported Frequency Outcome			Study Duration, Duration of Each Session, Number of Interventions, Intervention Intensity	tDCS Lead Locations		
			Monthly Migraine Days	Monthly Migraine Attacks	Headache Days Per Month		Cathode	Anode	tDCS Type (Cathodal vs. Anodal)
Antal, 2011 [29]	Germany	n = 26, 88%, 33 years	X	X		3 weeks, 15 min, 3 times per week, 1 mA	Oz (Primary visual cortex)	Cz (vertex)	Cathodal tDCS
Auvichayapat, 2012 [30]	Thailand	n = 37, 70%, 32 years		X		3 weeks, 20 min, 20 consecutive days, 1 mA	contralateral supraorbital	M1 (C3)	Anodal tDCS
Rocha, 2014 [31]	Brazil	n = 15, 97%, 24 years		X		3 weeks, 20 min, 3 times per week, 2 mA	Oz (Primary visual cortex)	Cz (vertex)	Cathodal tDCS
Ahdab, 2019 * [20]	Lebanon	n = 42, 83%, 36 years	X			1 week (2 weeks outcome measure), 20 min, 3 consecutive days, 2 mA	occipital cortex (O1/O2)	opposite supraorbital	Cathodal tDCS
Pohl, 2020 [32]	Switzerland	n = 23, 95%, 37 years	X		X	4 weeks, 20 min, 28 consecutive days, 1 mA	Cz (vertex)	Oz (Primary visual cortex)	Anodal tDCS
Sirin, 2021 [33]	Turkey	n = 29, 59%, 41 years		X	X	1 week (4 weeks outcome measure), 20 min, 3 consecutive days, 2 mA	contralateral supraorbital	M1 (C3)	Anodal tDCS
DaSilva, 2023 [28] #	USA	n = 25, 88%, 30 years			X	2 weeks, 20 min, 10 consecutive days, 2 mA	FC3 and FC5	C3 and C5	Both anodal and cathodal electrodes, centered on M1
Hasırcı Bayır, 2024 [27] #	Turkey	n = 58 (n = 22 with non-menstrual migraine) **, 100%, 30 years		X	X	1 week (4 weeks outcome measure), 20 min, 3 consecutive days, 2 mA	contralateral supraorbital	left dorsolateral prefrontal cortex (F3)	Anodal tDCS

* Cross-over design, ** This study included both menstrual-related and non-menstrual migraine. In the table, we only included data from participants with non-menstrual migraine, # These studies were excluded from the analysis (reasons discussed in the manuscript).

## Data Availability

All data are already available in the published paper on which this review is based. In response to a valid request, the corresponding author will provide data.

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
