# Peer review of "Transcranial Direct Current Stimulation in Episodic Migraine: A Systematic Review and Meta-Analysis of Randomized Controlled Trials†"

_medsci, 2025, doi:10.3390/medsci13030084_

Round 1
Reviewer 1 Report
Comments and Suggestions for Authors
This study evaluated the efficacy of tDCS compared to sham in preventing episodic migraine in adults. PubMed and Embase databases were searched until May 2025 to identify randomized controlled trials comparing tDCS with sham for the prevention of episodic migraine in adults. Risk of bias in the included trials was assessed using the Cochrane Risk of Bias Tool version 2. A random effect meta-analysis was conducted to evaluate the effects of cathodal and anodal tDCS on migraine frequency (days per month and attacks per month). The meta-analysis included six trials with 172 participants (mean age 34 years, 82% females). Both cathodal (three studies, over the occipital area) and anodal (three studies, over the occipital or primary motor area) tDCS reduced the mean number of monthly migraine days and migraine attacks compared to sham. After pooling the outcomes and excluding two studies at high risk of bias, anodal tDCS over the occipital or primary motor area and cathodal tDCS over the occipital area reduced headache frequency compared to sham. However, the reduction with anodal tDCS was not statistically significant. The authors cocluded that tDCS may be effective in preventing episodic migraine.
The article is well written, well-argued, and supported by data; however, some points require further clarification and improvement:
Line 86:
While many readers are familiar with the term placebo, the term sham may not be as widely understood. It is recommended to include a brief definition in the Methods section. For example:
“In clinical studies, particularly those involving physical or procedural interventions, sham treatments are used as a control condition to mimic the active intervention without delivering its therapeutic component. This design helps isolate the specific effects of the intervention from nonspecific influences such as the placebo effect, which includes psychological or physiological responses to receiving treatment.”
Line 114:
The paragraph mentions the phrase "The outcomes included…" leaving the reader with the possibility that other dependent variables could have been involved in the study. It is suggested to be more concise, for example:
"The only dependent variables measured were the number of migraine-related days in a month and the frequency of migraine attacks, and a composite outcome combining both."
Line 115:
There is a punctuation error with two consecutive periods ("..") at the end of the paragraph. Please revise to ensure proper formatting.
Line 117:
It would be very helpful for the reader to clarify what type of units correspond to the mean (and therefore the standard deviation). It is understood that the dependent variable measures time, but what units does it have? Are they "days" or "weeks"?
Line 169:
In Figure 4, under the column titled "Difference in Means", the unit of measurement is unclear. To improve reader comprehension, please indicate whether the values represent days, weeks, or another unit.
Line 184:
Similarly, in Figure 5, the column "Standard Difference in Means" lacks a specified unit. Indicating the unit (e.g., days or weeks) would enhance clarity and interpretability.
Line 187:
The same recommendation applies to Figure 6. The column "Standard Difference in Means" should include the corresponding unit of measurement to avoid ambiguity.
Author Response
Comment 1: While many readers are familiar with the term placebo, the term sham may not be as widely understood. It is recommended to include a brief definition in the Methods section. For example:
“In clinical studies, particularly those involving physical or procedural interventions, sham treatments are used as a control condition to mimic the active intervention without delivering its therapeutic component. This design helps isolate the specific effects of the intervention from nonspecific influences such as the placebo effect, which includes psychological or physiological responses to receiving treatment.”
Response 1: Thank you for your comment. We have incorporated the suggested text into Table 1, within the Methods section. We added an asterisk (*) after the word “sham” in the table and provided the corresponding explanation in the table legend (below the table). Please let us know if this meets your expectations. Line 127-130
.................................................................
Comment 2: The paragraph mentions the phrase "The outcomes included…" leaving the reader with the possibility that other dependent variables could have been involved in the study. It is suggested to be more concise, for example:
"The only dependent variables measured were the number of migraine-related days in a month and the frequency of migraine attacks, and a composite outcome combining both."
Response 2: This has been corrected in the Methods section based on the comment. Line 152
.................................................................
Comment 3: There is a punctuation error with two consecutive periods ("..") at the end of the paragraph. Please revise to ensure proper formatting.
Response 3: This has been corrected in the Methods section based on the comment. Line 154
.................................................................
Comment 4: It would be very helpful for the reader to clarify what type of units correspond to the mean (and therefore the standard deviation). It is understood that the dependent variable measures time, but what units does it have? Are they "days" or "weeks"?
Response 4: This has been corrected in the Methods section based on the comment as following: 'Descriptive data were reported as mean (of days or attacks per month) ± standard deviation (SD) or number (%).' Line 156
.................................................................
Comment 5: In Figure 4, under the column titled "Difference in Means", the unit of measurement is unclear. To improve reader comprehension, please indicate whether the values represent days, weeks, or another unit.
Response 5: Thank you for your comment. In the first column of Figure 4 under the title 'Outcome', we have specified 'monthly migraine attacks' in the first section, which clarifies that the means presented in that section refer to 'migraine attacks'. The second section is labeled 'monthly migraine days', indicating that the means in that section refer to 'migraine days'. Please let us know if this clarification is sufficient or if you have any further suggestions.
.................................................................
Comment 6: Similarly, in Figure 5, the column "Standard Difference in Means" lacks a specified unit. Indicating the unit (e.g., days or weeks) would enhance clarity and interpretability.
Response 6: Thank you very much for this thoughtful comment. We understand the importance of ensuring clarity in the presentation of effect sizes. The column labelled "Standard Difference in Means" refers to standardized mean differences (SMDs), which are commonly used in meta-analyses and are inherently unitless, as they reflect differences in terms of standard deviations rather than absolute units (e.g., days or weeks).
To avoid any confusion for readers, we have updated the figure legend to explicitly state that the values are standardized and do not carry a specific unit as the following: 'Standardized mean difference is a unitless measure that expresses the size of the effect in terms of standard deviations. It allows comparison across studies even when the frequency outcomes were measured using different scales. A higher absolute value indicates a stronger effect.' Line 230-233
We appreciate your suggestion, which helped improve the clarity of the figure.
.................................................................
Comment 7: The same recommendation applies to Figure 6. The column "Standard Difference in Means" should include the corresponding unit of measurement to avoid ambiguity.
Response 7: Please refer to comment 6. Line 237-239
………………………….
Reviewer 2 Report
Comments and Suggestions for Authors
Authors presented an interesting review about an original topic. Since the huge variability of the parameters used for non-invasive stimulation techniques, more researches regarding the topic are needed to have clarity on their effects on different disorders. I have just a few minor correction that I believe could improve the quality of the manuscript.
INTRO
Line 48: the term aura should be explained for non-headache specialists.
Line 56: “blood pressure lowering medications” are NOT migraine preventatives. This is a serious mistake from the authors. Just beta blockers and sartans are accepted as preventive therapies but certainly NOT for their BP lowering mechanism. There are different underlying mechanisms for each category, but BP is not one of them.
Line 60: citing the “limitations” of pharmacological intervention, the authors may give some example, such as side effects.
Line 61: better cite TMS in general, not just single pulse TMS (which is more intended as acute abortive therapy than preventive). The preventive one is the rTMS.
Among the non pharmacological intervention for migraine prevention, authors may cite lifestyle ones, such as Telecoaching for physical exercise (see doi: 10.3390/jcm14030861).
Line 66 authors may use the expression “non-invasive brain stimulation (NIBS) techniques”.
Line 68: better change “low-level” with “low-intensity”.
Line 69-72: authors should better explain tDCS functioning. They should cite the extracephalic or intracephalic montages, as well as the single or dual stimulation. In addition it should be mentioned the online (e.g., stimulation during cognitive tasks) or offline protocols. Lastly tACS and tRNS should be mentioned as well. In this scenario with so mani variables a systematic review would give a useful general view of best approaches.
RESULTS
Line 134: authors state they finally found 9 RCTs but the figure states 8. In addition, 2 were then excluded, giving a final result of 6 studies (which is consistent with 8 initial RCTs, not 9).
Author Response
INTRO
Comment 1: the term aura should be explained for non-headache specialists.
Response 1: Thank you for your comment. We revised introduction as the following: 'Migraine is classified into two types based on the presence or absence of aura: migraine with aura and migraine without aura [3]. Migraine with aura involves transient, fully reversible neurological symptoms—most commonly visual or sensory disturbances—that usually develop gradually over minutes and are typically followed by headache and other migraine-related features [3].' Line 54-57
......................................................
Comment 2: “blood pressure lowering medications” are NOT migraine preventatives. This is a serious mistake from the authors. Just beta blockers and sartans are accepted as preventive therapies but certainly NOT for their BP lowering mechanism. There are different underlying mechanisms for each category, but BP is not one of them.
Response 2: Thank you for your feedback. We agree that not all blood pressure-lowering medications are recommended for migraine prevention, and we did not intend to imply that blood pressure reduction is the main mechanism of action. Our original sentence was meant to list commonly used medication classes for migraine prevention, including antihypertensive agents which are supported by current guidelines.
To improve clarity, we have revised the sentence to:
“Some antihypertensive medications such as beta-blockers and sartans like candesartan, as well as anticonvulsants, tricyclic antidepressants, botulinum toxin, and anti-calcitonin gene-related peptide (anti-CGRP) medications are among the suggested medications for the prevention of migraine [6,8].” Line 65-66
In the original version, we used the broader and more reader-friendly term “blood pressure-lowering medications” instead of “antihypertensive medications,” aiming to make the text more accessible to a general audience. However, we have now adopted the more precise term “antihypertensive” to avoid any potential misunderstanding.
Please let us know if this revision adequately addresses your concern or if further clarification is needed.
......................................................
Comment 3: citing the “limitations” of pharmacological intervention, the authors may give some example, such as side effects.
Response 3: We revised the intro and added some limitations as suggested.
'Given the significant burden of migraine and the limitations of pharmacological treatments— such as adverse effects, lack of efficacy in some patients, and contraindications in others— non-invasive brain neuromodulation (NIBS) approaches have emerged as promising non-pharmacological alternatives for migraine prevention. These include transcranial magnetic stimulation (TMS), transcutaneous vagus nerve stimulation, transcutaneous supraorbital stimulation, and transcranial direct current stimulation (tDCS), some of which are already available in certain countries [7, 9, 10].' Line 70-71
......................................................
Comment 4: better cite TMS in general, not just single pulse TMS (which is more intended as acute abortive therapy than preventive). The preventive one is the rTMS.
Response 4: We revised the text accordingly. Please refer to the response to comment 3. Line 72-73
......................................................
Comment 5: Among the non pharmacological intervention for migraine prevention, authors may cite lifestyle ones, such as Telecoaching for physical exercise (see doi: 10.3390/jcm14030861).
Response 5: We added this section: 'In addition to neuromodulation, lifestyle-based interventions—such as telecoaching to manage physical activity— and behavioral treatments have also shown potential as non-pharmacological strategies for migraine prevention [7, 12].' Line 78-81
......................................................
Comment 6: authors may use the expression “non-invasive brain stimulation (NIBS) techniques”.
Response 6: We revised the text accordingly. Please refer to the response to comment 3. Line 71
......................................................
Comment 7: better change “low-level” with “low-intensity”.
Response 7: We revised the text accordingly. Line 82
......................................................
Comment 8: authors should better explain tDCS functioning. They should cite the extracephalic or intracephalic montages, as well as the single or dual stimulation. In addition it should be mentioned the online (e.g., stimulation during cognitive tasks) or offline protocols. Lastly tACS and tRNS should be mentioned as well. In this scenario with so mani variables a systematic review would give a useful general view of best approaches.
Response 8: Thank you for your comment. We revised the introduction accordingly.
'tDCS has two electrodes producing a constant low-intensity current flow from the anode (excitatory) to the cathode (inhibitory), altering cortical excitability by depolarizing or hyperpolarizing underlying neurons, respectively [13, 14]. Electrode placement can be cephalic (on the head) or extracephalic (for example, on the arms), and stimulation can be delivered in single or dual configurations, depending on the target area and study design [15-17]. Stimulation may also be applied during a task (online) or before it (offline), as the chosen protocol can influence clinical and behavioral outcomes [18]. Previous studies have reported that both cathodal and anodal tDCS are effective and safe for migraine prevention [19, 20]. ' Line 84-89
'Given the significant burden of migraine and the limitations of pharmacological treatments— such as adverse effects, lack of efficacy in some patients, and contraindications in others— non-invasive brain neuromodulation (NIBS) approaches have emerged as promising non-pharmacological alternatives for migraine prevention. These include transcranial magnetic stimulation (TMS), transcutaneous vagus nerve stimulation, transcutaneous supraorbital stimulation, transcranial alternating current stimulation (tACS), transcranial random noise stimulation (tRNS) and transcranial direct current stimulation (tDCS), some of which are already available in certain countries [7, 9, 10]. ' Line 72-76
......................................................
RESULTS
Line 134: authors state they finally found 9 RCTs but the figure states 8. In addition, 2 were then excluded, giving a final result of 6 studies (which is consistent with 8 initial RCTs, not 9).
Response 2: Thank you for highlighting this. We revised the restful section accordingly. Line 174
......................................................